# Study on Quality Change and Processing Suitability Evaluation of the Low-Temperature Vacuum Frying of Bananas

**DOI:** 10.3390/foods12091822

**Published:** 2023-04-27

**Authors:** Xi Chen, Yayuan Tang, Zhen Wei, Zhonglin Deng, Zhichun Li, Li Li, Xuemei He, Jian Sun

**Affiliations:** 1Guangxi Key Laboratory of Fruits and Vegetables Storage-Processing Technology, Guangxi Academy of Agricultural Sciences, Nanning 530007, China; pangdaxi2020@163.com (X.C.);; 2Guangxi South Subtropical Agricultural Science Research Institute, Guangxi Academy of Agricultural Sciences, Longzhou 532415, China; 3Agro-Food Science and Technology Research Institute, Guangxi Academy of Agricultural Sciences, Nanning 530007, China

**Keywords:** low-temperature vacuum-fried banana slices, factor analysis, evaluation model, SPME and GC×GC-TOFMS, flavor substances

## Abstract

The banana quality evaluation system is not sufficiently mature in China and cannot meet the demand of producing high-quality processed banana products. In order to screen banana varieties suitable for low-temperature vacuum frying and extend the banana deep processing industry chain, banana slices from 15 varieties planted in China were prepared by low-temperature vacuum-frying (VF) technology in the present study. After factor analysis on 20 indicators of sensory, flavor, nutritional and processing quality from different varieties of banana slices, comprehensive quality evaluation models were constructed for banana slices. It was concluded that Meishijiao No. 1 had the highest overall score among the 15 banana varieties; hence, it was deemed suitable for processing. Meanwhile, in order to investigate the difference between flavor substances in banana slices before and after processing, a flavor histology study was conducted with solid-phase microextraction (SPME) and comprehensive two-dimensional gas chromatography coupled to time-of flight mass spectrometry (GC×GC-TOFMS). It was found that the content differences of 2,3-pentanedione, hexanal and pentanal may cause the weakened fruitiness and the increased oil flavor of banana taste.

## 1. Introduction

Banana is one of the main crops in the tropical and subtropical regions [1]. Currently, more than 90% of bananas are eaten as fresh fruits. The average processing ratio of bananas is less than 5%, and the decay ratio is 25% [2]. Therefore, improving its processing ratio would be a critical means to developing the banana industry. Banana slices have been one of most popular processed foods worldwide because of their crispy taste, convenient transport and long storage period. Banana slices in the market are mainly from the Philippines, Vietnam and Thailand, and these banana slices are usually processed by high-temperature frying technology [3,4]. Banana slices processed by low-temperature vacuum frying (VF) technology can retain their original color, flavor and taste of fresh banana, and also have high nutritional and health value, by virtue of their low oil content, low acrylamide content, high anthocyanin content and high total carotenoids content [5,6,7]. However, there are only a few articles on vacuum-fried banana slice technology, and the research shows that the frying temperatures were above 100 °C [3,8]. The selection of suitable processing varieties for banana slices was also basically not involved. The above problems of banana slices have limited the development of the whole banana processing industry [2].

The quality of fruit slices is directly influenced by the variety of raw materials, maturity and processing technology [8,9]. Therefore, banana slices with unfavorable flavor quality are frequently found in the market. According to various research purposes, researchers have selected different indicators to evaluate the quality of banana slices. Thus far, the majority of studies have been conducted to establish evaluation models for banana drying [10,11]; for instance, mathematical models on the variation in the density and shrinkage of bananas during drying have been proposed [12]. Additionally, the effects of drying temperature, maturity and harvest time on the drying kinetics of banana products were also examined [13]. However, there are few studies that have comprehensively evaluate the quality of banana slices to select suitable varieties for banana slice processing.

The flavor of a banana slice is an indicator for evaluating the quality of banana slices. Generally, during the frying process of food, the formation mechanism and types of volatile metabolites are very complex and are related to the decomposition and interaction of substances inside the food, as well as the frying medium [14]. Bananas can release numerous aromatic components during ripening and greatly change in flavor after processing. It has been reported that different banana varieties have different aroma components, and these aroma components could be significantly changed during the postripening stage of fresh bananas [15,16,17]. Additionally, butyl acetate, butyl butyrate, hexyl acetate and amyl butyrate were identified as the main aroma components for the distinctive “fresh banana fruity” flavor, and these characteristic aroma components could be greatly changed by different processing methods [18]. In one study, 16 volatile components in banana slices dried by air-drying and vacuum microwave-drying methods were reduced and compared with those in fresh banana [19]. The changes in volatile components during the banana ripening and drying processes were analyzed in these studies. However, there are few reports on the change of aromatic components in banana slices induced by low-temperature vacuum frying.

Therefore, the objective of this study was to comprehensively screen suitable banana varieties for slice processing through building a quality evaluation model, as well as to identify flavor components of banana slices after low-temperature vacuum frying using the SPME and GC×GC-TOFMS method. The research findings can offer useful references in the development of the banana processing industry.

## 2. Materials and Methods

### 2.1. Materials

The sample information regarding the names and origins of the banana varieties are listed in Table 1. The fresh banana samples were free from diseases, insects and mechanical damage, and were soaked in ethephon solution for 3 min at a volume ratio of 400:1 (*w*/*v*), drained and ripened at 20 °C to the ripening stage 5 (the skin color of the bananas was yellow with a trace of green) [20].

### 2.2. Chemicals and Reagents

Chromatographic 2-octanol was obtained from Shanghai Yuanye Biotechnology Co., Ltd. (Shanghai, China), and 4,7-diphenyl-1,10-phenanthroline was obtained from Shanghai Yuanye Biotechnology Co., Ltd. (Shanghai, China). Folin phenol and a starch content detection kit were obtained from Beijing Solebar Technology Co., Ltd. (Beijing, China). Carbazole was obtained from Shanghai Maclean Biochemical Technology Co., Ltd. (Shanghai, China). Potassium hydrogen phthalate, trichloroacetic acid, phosphoric acid, ethanol, ferric chloride, ascorbic acid, sodium carbonate, gallic acid, phenol, concentrated sulfuric acid, sucrose, potassium sulfate, copper sulfate pentahydrate, boric acid, hydrochloric acid and galacturonic acid were obtained from Chengdu Kelong Chemical Co., Ltd. (Chengdu, China). All reagents were analytical reagents.

### 2.3. Sample Preparation

#### 2.3.1. Preparation of Fresh Banana Samples

Bananas at ripening stage 5 were washed, peeled and then cut into small pieces. Fresh banana pieces were put into liquid nitrogen for quick freezing, crushed into fresh banana powder at high speed, and then lyophilized (Pilot10-15 M, Biocool, Beijing, China) for 48 h to obtain dried fresh banana powder. The powder was stored at −20 °C.

#### 2.3.2. Preparation of Banana Slices by Low-Temperature Vacuum Frying

Bananas at the ripening stage 5 were washed, peeled, sliced (the thickness was 4 mm) and subjected to color protection with hardening for 5 min. Banana slices were obtained by low-temperature vacuum frying with a vacuum fryer (ZK-500, Xuzhong Machinery, Guangzhou, China). The vacuum degree was 0 MPa, and the frying temperature was 85 °C. The frying time and the centrifuge time were 50 min and 7 min, respectively. The banana slices were crushed in a high-speed mill (WND-500, Fei Libo, Shanghai, China), sieved by a 60-mesh sieve and vacuum-packed and frozen at −20 °C. All banana samples were processed in the same way. However, the samples for the determination of sensory quality index were banana slices, and they were not crushed.

### 2.4. Determination of the Sensory Quality Index

The color was measured with a spectrophotometer (CM-3600A, Konica minolta, Tokyo, Japan), and the results showed L*, a* and b* values, with L* representing lightness (L* = 0 denotes black, and L* = 100 denotes white), a* representing red (+) or green (−), and b* representing yellow (+) or blue (−). The L*, a* and b* parameters were can be read directly from the device [21]. Hardness and brittleness were measured with a texture tester (CT3, Brookfield, Middleboro, MA, USA), and the parameters are shown in Table 2 [22,23].

### 2.5. Determination of the Flavor Quality Index

Total soluble solid (TSS) was determined with a refractometer (PAL-1, Atago, Guangzhou, China) [24]. Soluble sugar (SS) content was determined via the phenol-sulfuric acid method, and titratable acid (TA) content was determined via acid–base indicator titration [25,26]. The contents were all demonstrated as %. The sugar–acid ratio was calculated as the ratio of the soluble sugar content to the titratable acid content for each sample.

### 2.6. Determination of the Nutritional Quality Index

Protein content was determined using the Kjeldahl method, and a digestion furnace (KDN-08C, Tuopuyunnong, Hangzhou, China) and Kjeldahl azotometer (KDN, Tuopu, Hangzhou, China) were used, with the results presented as % [27]. Starch content was determined using visible spectrophotometry (BioTek, Winooski, VT, USA) with a Solarbio starch content detection kit (Solarbio, Beijing, China), and the testing method was carried out according to the instructions of this kit. The content is expressed as mg/g. The crude fiber content was washed and dried with fiber tester (F800, Haineng, Dezhou, China) and a box resistance furnace (SX2-8-10A, Jiecheng, Shanghai, China) according to a previously described method [28], with the results presented as %. Pectin content was determined using the spectrometric detection method with a microplate reader (BioTek, Winooski, VT, USA) [29], with the content express as g/kg. Fat content was determined using the soxhlet extraction method [9,30], with the content expressed as %. Total phenolic content was determined with folin–phenol colorimetry [31], with the content expressed as mg/g. Vitamin C (Vc) content was determined using spectrophotometry (BioTek, Winooski, VT, USA) according to a previously described method [29], with the content expressed as mg/100 g.

### 2.7. Determination of the Processing Quality Index

The degree of expansion was the ratio of banana slice thickness after frying to that before frying, with 10 replicates for each banana sample. The output ratio was the ratio of the quality of banana slices after frying to that before frying (after slicing and before pretreatment). The calculation method of rehydration was the ratio of the weight of banana slices after (they were drained for 5 min after soaking for 30 min) to before soaking. Moisture content was determined with a halogen moisture meter (XFSFY-50A, Xiongfa, Xiamen, China).

### 2.8. Determination of Differential Flavor Substances in Bananas before and after Low-Temperature Vacuum Frying

The SPME—GC×GC-TOFMS method was used to determine flavor substances in the banana [32,33,34]. A 1 g sample was put into a 20 mL headspace bottle. Then, 10 μL of 40 μg/mL 2-octanol was added into the bottle as an internal standard, placed at 60 °C for 10 min and then extracted for 40 min. The 2 cm DVB/CAR/PDMS extraction fiber (Supelco, Bellefonte, PA, USA) was inserted into the headspace of the bottle to adsorb for 5 min.

GC×GC-TOFMS was performed using a comprehensive two-dimensional gas chromatography (7890A, Agilent, Santa Clara, CA, USA) coupled to a time-of-flight mass spectrometer (Pegasus 4D, LECO, St. Joseph, MI, USA) with a TG-WAXMS column (30 m × 250 μm × 0.25 μm, Agilent, Santa Clara, CA, USA) and a DB-17MS column (2 m × 100 μm × 0.10 μm, Agilent, Santa Clara, CA, USA) used as the second column. Helium (purity > 99.9999%) was used as the carrier gas at a flow rate of 1.0 mL/min, and the injector split ratio of the split/splitless injector was set to 1:5. The injection temperature was 250 °C. The initial oven temperature was 40 °C, and this temperature was held for 5 min, then raised to 70 °C at 2 °C/min, raised to 250 °C at 8 °C/min and then held for 7.5 min. The second oven was always 15 °C hotter than the first oven. The modulation period was 4.0 s, and a modulator temperature offset of 15 °C above the second oven was applied. The injector port was heated to 270 °C, and the ion source was maintained at 230 °C. The mass range was 20–450 *m/z*, and ionization energy was 70 eV. The voltage for the detector was 1680 V. The acquisition rate was 50 spectra/s. Mass spectra were probability matched to the NIST (National Institute of Standards and Technology) 2014 Mass Spectral Library (Gaithersburg, Maryland, MD, USA).

### 2.9. Statistical Analysis

The factor analysis method was used to analyze the quality indexes of banana slices. All the experiment were repeated at least three times, and the results are expressed as means with standard deviations. SPSS (IBM Co. Chicago, IL, USA, version 25.0) software and R (Lucent Technologies, Paris, France, version 3.3.2) software were used for data analysis.

## 3. Results and discussion

### 3.1. Analysis on the Comprehensive Quality of Banana Slices

#### 3.1.1. Analysis on Sensory Quality of Banana Slices

The evaluation of the sensory quality of the banana slices is shown in Table 3. Only the coefficient of variation of the L* value was less than 10%, indicating that there was little difference in the brightness index of color among the different banana slices. The coefficients of variation of the other four sensory quality indicators were all over 20%, indicating that there was large difference in these data among the different varieties of banana slices. Crispness had the highest coefficient of variation, followed by hardness and color a* and b*. These four indexes had great influence on the sensory quality of banana slices and were representative of the sensory evaluation of banana slices. The indicators that had significant impact on quality were selected for subsequent analysis. Therefore, hardness, brittleness, and the color a* and b* were selected for the subsequent development and analysis of a quality evaluation model.

#### 3.1.2. Analysis of the Flavor Quality of Banana Slices

The horizontal analysis of the flavor quality of the banana slices is shown in Table 4. The coefficients of variation for SS content, TA content and sugar–acid ratios were all greater than 10%, while that of TSS content was not. These indicators had a deeper impact on flavor quality because they had a significant degree of data dispersion. The sugar–acid ratio was positively and negatively correlated with SS content and TA content, respectively. Generally, fruit flavor was closely related to its sugar and acid levels. Therefore, the sugar–acid ratio was chosen for the subsequent analysis.

#### 3.1.3. Analysis on the Nutritional Quality of Banana Slices

The horizontal analysis of the nutritional quality of the banana slices is shown in Table 5. The coefficients of variation of nutritional indicators other than fat content were greater than 10%, indicating that these data were highly dispersed. Of these, starch content had the largest coefficient of variation, followed by crude fiber, phenolics, pectin, protein and Vc contents. This indicated that variability of varietal and origin had a considerable influence on the nutritional quality in banana slices. Therefore, these indicators were selected for the subsequent analysis.

#### 3.1.4. Analysis on Processing the Quality of Banana Slices

As shown in Table 6, the horizontal analysis of banana slice processing quality indicators showed that the coefficient of variation of the rehydration ratio was the smallest among them. The degree of expansion and moisture content were all above 10%, with a high degree of data dispersion, and the moisture content changed most significantly among all indicators. This indicated that there were major differences in the processing indicators of different banana slices. Therefore, the degree of expansion, output ratio and moisture were selected for the development and analysis of a quality evaluation model.

### 3.2. Screening of Core the Indexes of Banana Slices

The complexity of the indicators of banana slices makes analysis difficult. Factor analysis was applied to obtain a few common factors, to explain the variables from the multiple indicators of banana slices in this paper and finally to achieve the idea of dimensionality reduction. Factor analysis was carried out on 14 selected banana slices indicators using SPSS software. The results were shown in Table 7. The analysis revealed that there were 5 common factors with eigenvalues greater than 1; the cumulative percent of variance was 84.322%, and they could reflect most of the indicators well. The correlation analysis of the different banana slice quality indicators is shown in Table 8. The percent of variance for common factor 1 was 26.827%, with a focus on brittleness, hardness, degree of expansion, protein, starch and output ratio information in order of factor loadings. Hardness, degree of expansion, protein and brittleness were highly significantly correlated (R values, 0.873 **, 0.800 **, −0.658 **). Starch was significantly related to brittleness (R = 0.584 *), and brittleness had the highest factor loadings among them, while the output ratio was important in practical production. Therefore, the brittleness and output ratio were determined to be the representative indicators for the common factor 1.

The percent of variance for common factor 2 was 18.748%. It gathered information on pectin and b* and a* values indicating color, output ratio, and polyphenols. Polyphenols were significantly correlated with the output ratio (R = −0.547 *), overlapping with common factor 1. The a* value was significantly correlated with b* values (R = 0.524 *), and the b* value had the largest factor loadings. The b* value has practical significance because it can indicate the degree of blue–yellow in banana slices. Therefore, b* values and pectin were chosen as the representative indicators for common factor 2.

In common factor 3, the percent of variance was 14.015% and mainly included information on sugar–acid ratio and starch content. The contribution of the sugar–acid ratio was much greater than that of starch, which overlapped with the information on factor 1. Therefore, the sugar–acid ratio was chosen as a representative indicator for the common factor 3.

The percent of variance for common factor 4 was 13.232% and was mainly related to Vc and crude fiber contents. Despite the component information Vc factor loadings reaching as high as 0.929, the Vc to output ratio was highly significantly correlated (R = −0.691 **), and the output ratio indicator in the common factor 1 was duplicated. Therefore, crude fiber content was the representative indicator for common factor 4.

The percent of variance for common factor 5 was 11.500% and consisted mainly of moisture information with factor loadings up to 0.922. Therefore, moisture was a representative indicator of common factor 5. In summary, b* value, brittleness, sugar–acid ratio, crude fiber, pectin, output ratio and moisture were finally selected as the core indicators for the comprehensive quality evaluation of banana slices, which basically cover the sensory, flavor, nutrition and processing quality of banana slices.

### 3.3. Development of a Comprehensive Quality Evaluation Model for Banana Slices

With the sensory, flavor, nutritional and processing quality index matrix of banana slices taken as variables, the eigenvalues of common factor, percent of variance and factor loadings of each variable in the corresponding common factor were obtained using factor analysis. Based on the research methods of Jing et al. [35], the weight value of each index was obtained with Formula 1, and then the seven core indicators for evaluating banana slices were normalized to obtain their corresponding weight values, as listed in Table 9. Therefore, the evaluation model of comprehensive quality was as follows: F_comprehensive score_ = b* value × 0.1742 + brittleness × 0.2201 + sugar-acid ratio × 0.2724 + crude fiber × 0.2093 − pectin × 0.1668 + output ratio×0.1220 + moisture × 0.1689.
(1)$$Q_{n=}\sum_{j=1}^{m}\frac{Z_{jn}}{\sqrt{\lambda_{j}}}\times E_{j}$$
where: *Q_n_* is the weight value of each index calculated by factor analysis, *Z_jn_* is the factor loadings of the *j*-th common factor of the nth index, *λ_j_* is the eigenvalue of the *j*-th common factor, *E_j_* is the ratio of the percent of variance to the cumulative percent of variance for the jth common factor, and m is the number of common factors with eigenvalues greater than 1, with a fixed value 5.

To avoid the impact of the different measurement units and data scales for the core indexes of banana slices, the above indexes were converted for data normalization. Among them, the brittleness and moisture of banana slices were negative indicators, with a lower value indicating the better quality of banana slices. The other indexes were positive indicators because a greater proportion of them had better quality. The data of each index were converted according to Formulas (2) and (3). The standardized data of the core indicators of banana slices are shown in Table 10.
(2)$$\mathrm{negative}\;\mathrm{indicator}=\frac{{maxF}_{xy}{-F}_{xy}}{{maxF}_{xy}-{minF}_{xy}}$$
(3)$$\mathrm{positive}\;\mathrm{indicator}=\frac{F_{xy}-{minF}_{xy}}{{maxF}_{xy}-{minF}_{xy}}$$
where *F_xy_* is the original measured value of the *y*-th index for the *x*-th banana slices.

The standardized data on the core indexes were substituted into the evaluation model of comprehensive quality to obtain comprehensive scores of all banana slices. Qualitative analysis was then carried out by means of systematic cluster analysis. The suitability of banana slices for processing was divided into five categories, as detailed in Table 11: class I (very suitable), class II (more suitable), class III (basically suitable), class IV (less suitable) and class V (very unsuitable). Table 11 shows that the quality of banana slices varied considerably between varieties, with the range of variation being 0.2966–0.7550. Among them, Meishijiao No. 1 had the highest overall score and ranked first, followed by Guijiao No. 6 and Guijiao No. 2, which were very suitable for processing banana slices. Therefore, Meishijiao No. 1 was used for the following analysis of differential flavor substances.

A variety of domestically grown bananas were selected, dimensionality reduction analysis was performed on 14 quality indicators through factor analysis, 7 core indicators were obtained, and these covered the quality of banana slices such as color and taste. These indicators can comprehensively reflect the comprehensive quality of slices and can predict the processing suitability of unknown banana varieties. Thus far, little research has been conducted on the suitability of banana processing; rather, experts from China and other countries have conducted suitability studies on other crops, such as pear [36], peach [37], citrus [38], sweet pepper [39], apple and [40] potato [41]. The quality of different fruits and vegetables was analyzed in these studies, and only those varieties mentioned in these texts were comprehensively evaluated, but not the unknown samples.

### 3.4. Qualitative Analysis of Differential Flavor Substances in Banana before and after Low-Temperature Vacuum Frying

The formation of banana aroma substances is closely related to its variety, cultivation environment, maturity and storage method [16,42,43]. There was a great difference in volatile components of bananas before and after maturity. The aroma and taste of bananas before and after low-temperature vacuum frying were also different, with the banana fruit flavor decreasing and fat flavor increasing. To determine the key substances causing this difference, 79 different flavor substances were found through chromatographic analysis, as shown in Table 12, including 29 aldehydes, 15 ketones, 12 alcohols, 7 esters, 4 acids, 3 furans, 2 ethers and 1 phenol. Aldehydes and alcohols are the main volatile substances in the banana growing stage, while esters and alcohols are the main aroma components [15,44].

### 3.5. OPLS-DA of Differential Flavor Substances in Banana

Orthogonal projections to latent structures discriminant analysis (OPLS-DA) was carried out based on the obtained data to reveal the distribution of the samples, the score of which plots are shown in Figure 1. The first principal component (PC1) and the second principal component (PC2) accounted for 76.1% and 9.8% of the variance, respectively, explaining 85.9% of the variance. The samples were clearly separated in the OPLS-DA score plots. The score of the VF group was in the negative area of PC1, while the score of the fresh fruit (XG) group before VF was in the positive area of PC1. This showed that there was a significant difference in the distribution of banana flavor substances before and after VF. The results showed that the changes of metabolic pathway of banana after VF resulted in the difference of volatile metabolites.

In OPLS-DA, the R^2^X was 0.929, the R^2^Y was 0.999, and the Q^2^ was 0.932. The predictability of the model and the interpretability of independent variable and dependent variable were higher than 0.900. This indicated that the model was stable and the data were reliable. OPLS-DA could reduce the complexity and improve the interpretation ability of the model, and it maximized the differences between data groups. However, it could lead to the overfitting problem of the OPLS-DA model. Therefore, the data needed to be replaced to evaluate the current model fitting phenomenon, with the specific replacement inspection diagram being seen in Figure 2, where the R^2^ intercept value is (0.0, 0.91) and Q^2^ intercept value is (0.0, 0.25). All the blue points on the left are lower than the original blue points on the far right of Q^2^, indicating that OPLS-DA model had no overfitting and was an effective model, which could be used to screen different flavor substances in this study.

### 3.6. Identification of Key Differential Flavor Substances in Bananas

Generally, VIP could be used to measure the influence of the expression pattern of each metabolite on the classification and discrimination of each group of samples and the explanatory power. According to the VIP size, the variable (characteristic peak) could explain the importance of the X dataset and the associated Y dataset. The variable with VIP > 1 was used as one of the screening conditions for potential biomarkers. As shown in Figure 3, 16 substances with VIP > 1 were identified, including 1 ether, 1 acid, 3 alcohols, 4 ketones and 7 aldehydes. The content changes of 10 substances before and after VF were significantly different (*p* < 0.01), the change range of VIP was 1.03~5.31, which indicated that the substances changed significantly before and after VF of banana, and had an important impact on the flavor change of bananas before and after processing.

The heatmap and cluster analysis of key different flavor substances in bananas before and after VF are shown in Figure 4. It can be seen that 16 substances were divided into two categories, only 2-Hexenal was reduced in relative content after VF and was thus separated into a large category. The relative content of other metabolites increased, and the combination was another major category.

### 3.7. Identification of the Substances Affecting the Sensory Flavor in Bananas before and after Low-Temperature Vacuum Frying

By comparing the 79 obtained compounds described in Section 3.4 with Flavor DB, we found that 12 metabolites play a major role in representing 10 different sensory flavors, such as sweetness, citrus, fruitiness and pungent smell. These compounds included 1 ester, 2 alcohols, 4 ketones and 5 aldehydes, as shown in Figure 5. Then, the sensory flavor of bananas before and after VF was compared and analyzed, and the details are provided in Figure 6. Before and after VF, the flavor of fat and fruit in the bananas mostly changed, which was followed by a pungent, fatty, aldehydic, fresh, ethereal, sweet, green and citrus flavor according to the analysis of the different flavor substances before and after the VF described in Section 3.6. Among the 12 substances that affect the sensory flavor, 2,3-pentanedione, hexanal and pentanal had significant differences before and after VF. It could be inferred that the contents of these three substances changed significantly before and after VF in bananas, which led to a greater flavor change before and after VF, such as the flavor of fattiness and fruitiness. Among them, 2,3-pentanedione, as a food additive, had a sweet taste like butter and honey, which was consistent with the sensory flavor associated with that in this paper. Meanwhile, hexanal was involved in a variety of sensory flavors, such as fruity, fatty and green flavors. Pentanal was associate with a fruity and pungent flavor. The significant difference in the content of these substances may be the main reason for the significant change of banana flavor before and after VF.

Many studies have been conducted on the changes in volatile components during banana ripening and on the volatile components in banana powder [45,46], banana wine [47] and banana vinegar [48]. However, few studies have sought to identify the volatile components before and after frying banana slices. This study lays a foundation for improving the quality of the banana slices.

## 4. Conclusions

In this study, the comprehensive quality of banana slices was evaluated using factor analysis, and the evaluation model for the comprehensive quality of banana slices was established. The most suitable banana variety for processing was Meishijiao No. 1, followed by Guijiao No. 6 and Guijiao No. 2. Through the GC×GC-TOFMS method, the change of banana flavor composition before and after VF was explored, and it was found that the key substances affecting the sensory flavor of VF bananas might be 2,3-pentanedione, hexanal and pentanal. This study offers a theoretical foundation for the adjustment of banana sensory flavor and the optimization of VF process conditions. Further studies on the mechanism of flavor change of banana before and after frying should be pursued.

## Figures and Tables

**Figure 1 foods-12-01822-f001:**
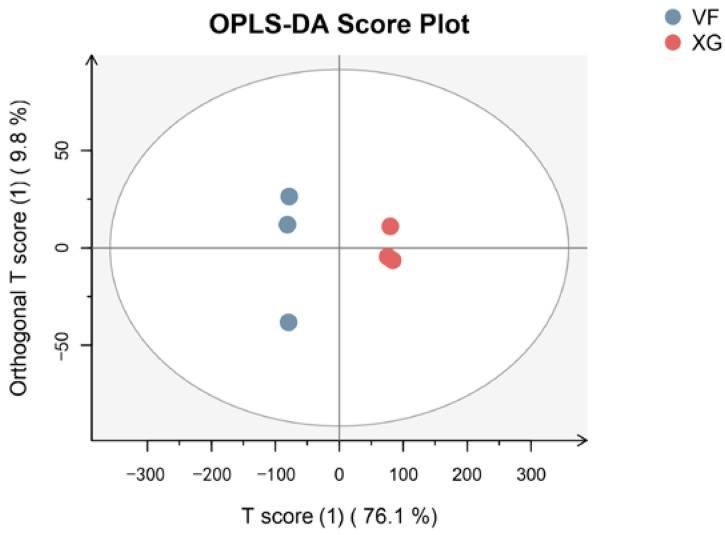
Analysis of the flavor substances of the OPLS-DA of banana before and after VF.

**Figure 2 foods-12-01822-f002:**
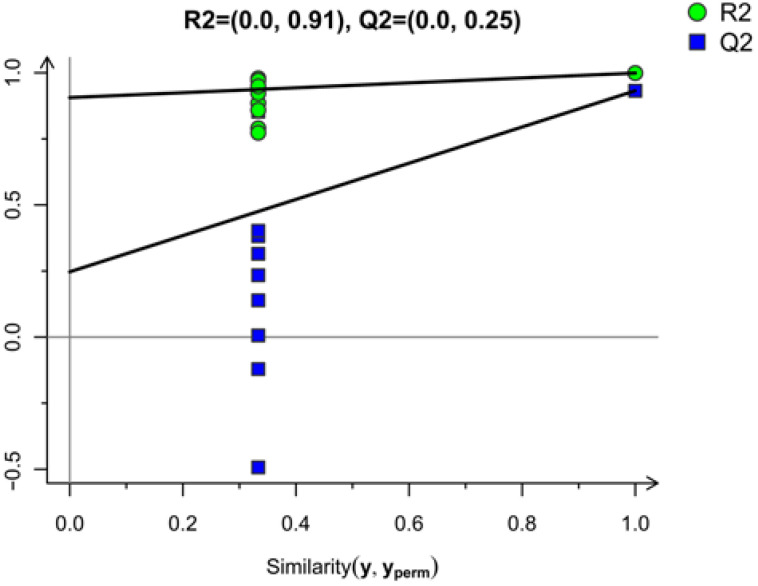
OPLS-DA test chart for banana flavor before and after VF.

**Figure 3 foods-12-01822-f003:**
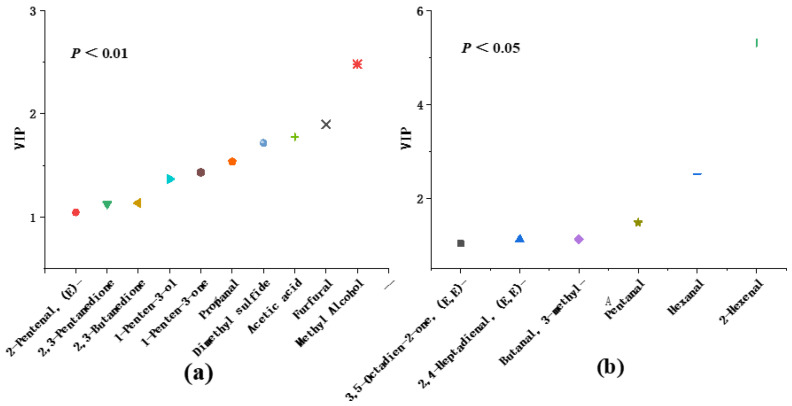
Comparison of VIP values of key flavor substances in bananas before and after VF. (**a**) The key flavor substances in bananas before and after VF (*p* < 0.01). (**b**) The key flavor substances in bananas before and after VF (*p* < 0.05).

**Figure 4 foods-12-01822-f004:**
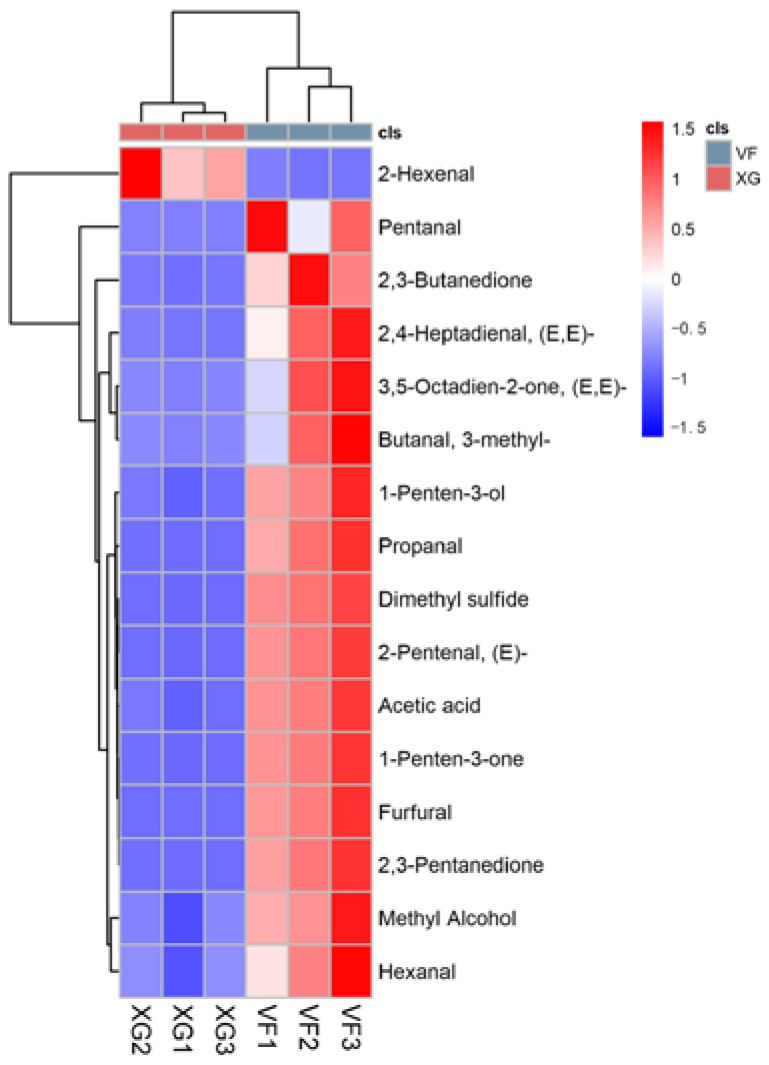
Heatmap and cluster analysis of the key flavor substances in bananas before and after VF.

**Figure 5 foods-12-01822-f005:**
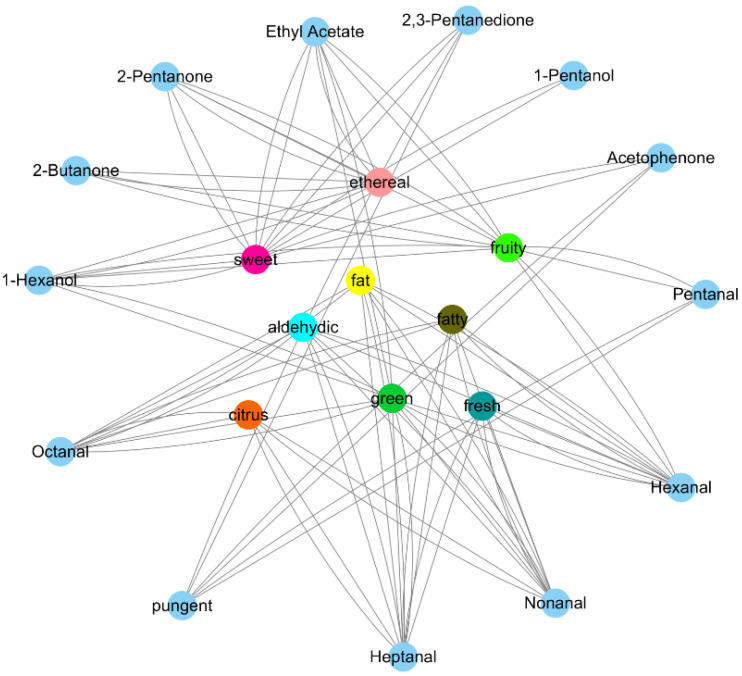
Correlation network between banana flavor substances and sensory flavor before and after VF.

**Figure 6 foods-12-01822-f006:**
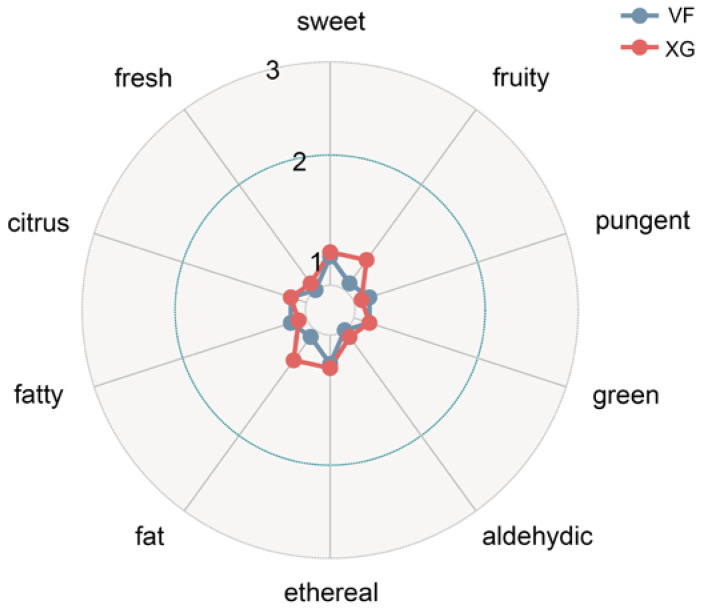
Radar chart of correlation between banana flavor substances and sensory flavor before and after VF.

**Table 1 foods-12-01822-t001:** Name and origin of the banana raw materials.

No.	Name	Origin
1	Guijiao No. 1	Longan, Guangxi
2	Guijiao No. 2	Honghe, Yunnan
3	Guijiao No. 6	Honghe, Yunnan
4	Guizao	Honghe, Yunnan
5	Zhongjiao No. 9	Honghe, Yunnan
6	Reke No. 1	Haikou, Hainan
7	Reke No. 2	Haikou, Hainan
8	Reke No. 4	Haikou, Hainan
9	Baodao	Honghe, Yunnan
10	Baxi	Honghe, Yunnan
11	Jinfen No. 1	Honghe, Yunnan
12	Fenjiao	Longan, Guangxi
13	Meishijiao No. 1	Zhanjiang, Guangdong
14	151	Zhanjiang, Guangdong
15	CB5	Zhanjiang, Guangdong

Note: Bananas with the codes “151” and “CB5” were germplasm and kept in the Resource Garden of the Institute of Fruit Tree Research, Guangdong Academy of Agricultural Sciences.

**Table 2 foods-12-01822-t002:** Setting parameters of the hardness and brittleness.

Parameters	Hardness	Brittleness
Type of the probe	TA 44	TA 7
Trigger force (g)	5	5
Test distance (mm)	2.5	7
Test speed (mm/s)	2	1.5

**Table 3 foods-12-01822-t003:** Horizontal analysis on sensory quality indexes of banana slices.

Index	Amplitude	Mean Value	Standard Deviation	Coefficient of Variation (%)
L* value	62.35–76.49	69.99	3.96	5.66
a* value	2.89–8.67	4.76	1.55	32.50
b* value	23.33–48.72	29.93	6.03	20.16
hardness (g)	1397.00–4421.70	2414.04	921.57	38.17
brittleness (g)	941.50–4115.33	1943.93	885.41	45.55

Note: The coefficient of variation was the ratio of standard deviation to mean value, as below. L*, a* and b* values represent color, refer to Section 2.4. for details, * has no special meaning among them, as below.

**Table 4 foods-12-01822-t004:** Horizontal analysis on flavor quality indexes of banana slices.

Index	Amplitude	Mean Value	Standard Deviation	Coefficient of Variation (%)
TSS (%)	5.60–7.80	6.89	0.66	9.55
SS (%)	32.27–66.23	49.65	9.22	18.58
TA (%)	1.15–2.16	1.49	0.25	17.04
sugar-acid ratio	17.70–45.67	34.08	7.48	21.94

Note: TSS—total soluble solid; SS—soluble sugar; TA—titratable acid.

**Table 5 foods-12-01822-t005:** Horizontal analysis on the nutritional quality indexes of banana slices.

Index	Amplitude	Mean Value	Standard Deviation	Coefficient of Variation (%)
Protein (%)	2.52–5.84	3.88	0.99	25.63
Starch (mg/g)	59.44–392.48	203.16	104.89	51.63
Crude fiber (%)	0.96–3.33	1.84	0.67	36.70
Pectin (g/kg)	10.85–28.47	19.10	5.85	30.66
Fat (%)	15.84–24.67	20.48	2.03	9.92
Phenolics (mg/g)	0.76–4.11	2.66	0.94	35.21
Vitamin C (mg/100 g)	25.40–52.53	38.17	9.47	24.80

**Table 6 foods-12-01822-t006:** Horizontal analysis on the processing quality indexes of banana slices.

Index	Amplitude	Mean Value	Standard Deviation	Coefficient of Variation (%)
Degree of expansion	0.87–1.42	1.12	0.16	14.53
Output ratio (%)	22.34–39.95	29.06	5.18	17.83
Rehydration ratio	1.53–2.16	1.80	0.18	9.90
Moisture (%)	1.68–4.03	3.07	0.77	24.96

**Table 7 foods-12-01822-t007:** Rotated component matrix for the quality index of banana slices.

Index	Common Factor				
	1	2	3	4	5
a* value	0.044	0.761	0.419	−0.067	0.120
b* value	0.096	0.795	−0.030	−0.323	0.166
Hardness	0.879	0.167	0.010	−0.058	−0.134
Brittleness	0.955	−0.110	−0.025	−0.068	0.020
Sugar–acid ratio	0.038	0.088	0.926	0.223	0.049
Protein	−0.757	−0.254	0.315	−0.192	0.390
Starch	0.600	0.039	−0.736	−0.005	−0.007
Crude fiber	0.082	0.047	0.448	0.653	−0.240
Pectin	0.069	−0.805	0.026	−0.231	0.233
Phenolics	−0.079	−0.552	0.209	0.162	0.595
Vc	−0.188	−0.151	0.023	0.929	0.206
Degree of expansion	0.884	−0.046	0.027	−0.168	0.150
Output ratio	0.538	0.557	−0.193	−0.501	−0.119
Moisture	−0.054	0.096	−0.051	0.011	0.922
Eigenvalues (λ)	3.756	2.625	1.962	1.853	1.610
Percent of variance (%)	26.827	18.748	14.015	13.232	11.500
Cumulative percent of Variance (%)	26.827	45.575	59.590	72.822	84.322

**Table 8 foods-12-01822-t008:** Correlation analysis on the quality indexes of banana slices.

Index	a* Value	b* Value	Hardness	Brittleness	Sugar-Acid Ratio	Protein	Starch	Crude Fiber	Pectin	Polyphenols	Vc	Degree of Expansion	Output Ratio	Moisture
a* value	1													
b* value	0.524 *	1												
Hardness	0.249	0.057	1											
Brittleness	0.021	−0.006	0.873 **	1										
Sugar–acid ratio	0.473	−0.074	0.094	−0.038	1									
Protein	−0.027	−0.192	−0.712 **	−0.658 **	0.203	1								
Starch	−0.176	0.072	0.550 *	0.584 *	−0.599 *	−0.681 **	1							
Crude fiber	0.019	−0.091	−0.113	−0.018	0.477	−0.132	−0.308	1						
Pectin	−0.441	−0.521 *	−0.153	0.184	−0.075	0.281	0.061	−0.142	1					
Phenolics	−0.370	−0.256	−0.304	−0.084	0.167	0.419	−0.290	−0.008	0.392	1				
Vc	−0.099	−0.409	−0.248	−0.204	0.220	0.087	−0.137	0.494	−0.026	0.375	1			
Degree of expansion	−0.113	0.264	0.628 *	0.800 **	−0.031	−0.579 *	0.445	0.078	0.097	0.148	−0.327	1		
Output ratio	0.308	0.682 **	0.531 *	0.456	−0.254	−0.561 *	0.461	−0.187	−0.296	−0.547 *	−0.691 **	0.574 *	1	
Moisture	0.163	0.111	−0.123	−0.081	0.032	0.380	0.015	−0.191	0.182	0.327	0.171	0.036	−0.054	1

Note: * after the number indicates significant correlation, *p* < 0.05; ** indicates extremely significant correlation, *p* < 0.01.

**Table 9 foods-12-01822-t009:** Weight values on core indexes of banana slices.

	b* Value	Brittleness	Sugar–Acid Ratio	Crude Fiber	Pectin	Output Ratio	Moisture
Weight value	0.1742	0.2201	0.2724	0.2093	−0.1668	0.1220	0.1689

**Table 10 foods-12-01822-t010:** Data after standardized treatment on the core indexes of banana slices.

No.	Name	b* Value	Brittleness	Sugar–Acid Ratio	Crude Fiber	Pectin	Output Ratio	Moisture
1	Guijiao No. 1	0.19	0.77	0.93	0.17	0.23	0.19	0.53
2	Guijiao No. 2	0.18	1.00	0.66	0.76	0.12	0.12	0.65
3	Guijiao No. 6	0.28	0.86	0.86	0.67	0.34	0.07	0.94
4	Guizao	0.17	0.45	0.97	0.61	0.68	0.60	1.00
5	Zhongjiao No. 9	0.08	0.73	0.52	0.00	0.81	0.45	0.73
6	Reke No. 1	0.18	0.95	0.95	0.05	0.81	0.30	0.00
7	Reke No. 2	0.30	0.90	0.89	0.53	0.03	0.23	0.05
8	Reke No. 4	0.27	0.93	0.71	0.29	0.32	0.34	0.49
9	Baodao	0.19	0.88	0.85	0.24	0.75	0.00	0.05
10	Baxi	0.22	0.73	0.89	0.25	1.00	0.24	0.03
11	Jinfen No. 1	0.01	0.00	0.96	0.21	0.71	0.28	0.43
12	Fenjiao	0.00	0.53	0.92	1.00	0.78	0.29	0.17
13	Meishijiao No. 1	1.00	0.56	0.88	0.38	0.18	1.00	0.28
14	151	0.40	0.68	0.68	0.13	0.00	0.78	0.48
15	CB5	0.43	0.28	0.78	0.30	0.27	0.83	0.26

Note: All standardized data were obtained from the average value (repeated 3 times of) through standardization, as below.

**Table 11 foods-12-01822-t011:** Comprehensive quality order, comprehensive score and class of banana slices.

Name	Origin	Order	Score	Class
Meishijiao No. 1	Zhanjiang, Guangdong	1	0.7550	1
Guijiao No. 6	Honghe, Yunnan	2	0.7243	1
Guijiao No. 2	Honghe, Yunnan	3	0.6948	1
Guizao	Honghe, Yunnan	4	0.6481	2
Reke No. 2	Haikou, Hainan	5	0.6361	2
151	Zhanjiang, Guangdong	6	0.6088	2
Reke No. 4	Haikou, Hainan	7	0.5770	3
Guijiao No. 1	Longan, Guangxi	8	0.5641	3
CB5	Zhanjiang, Guangdong	9	0.5120	3
Fenjiao	Longan, Guangxi	10	0.5090	3
Reke No. 1	Haikou, Hainan	11	0.4089	4
Baodao	Honghe, Yunnan	12	0.3920	4
Baxi	Honghe, Yunnan	13	0.3625	4
Zhongjiao No. 9	Honghe, Yunnan	14	0.3603	4
Jinfen No. 1	Honghe, Yunnan	15	0.2966	5

**Table 12 foods-12-01822-t012:** Data after standardized treatment on core indexes of banana slices.

No.	Compound	CAS	RT	VIP	Fold Change	*p* Value
1	1-Butanol	71-36-3	14.87	0.18	36.08	0.02
2	1-Hexanol	111-27-3	25.20	0.15	0.47	0.60
3	1-Pentanol	71-41-0	21.13	0.72	13.09	0.06
4	1-Penten-3-ol	616-25-1	15.67	1.37	5.51	0.00
5	1-Penten-3-one	1629-58-9	9.07	1.43	15.00	0.00
6	1-Propanol	71-23-8	9.73	0.13	0.59	0.26
7	2-Butanone	78-93-3	5.60	0.76	11.39	0.00
8	2-Butenal, (E)-	123-73-9	11.13	0.18	49.03	0.15
9	2-Cyclohexen-1-one, 3,4,4-trimethyl-	17299-41-1	27.67	0.02	0.60	0.19
10	2-Heptenal, (Z)-	57266-86-1	24.67	0.85	33.83	0.02
11	2-Hexenal	505-57-7	19.67	5.31	0.04	0.01
12	2-Nonenal, (E)-	18829-56-6	29.87	0.04	1.16	0.32
13	2-Octenal, (E)-	2548-87-0	27.67	0.37	3.52	0.00
14	2-Octenal, 2-butyl-	13019-16-4	32.00	0.80	0.04	0.03
15	2-Pentanone	107-87-9	7.467	2.27	9.52	0.12
16	2-Penten-1-ol, (Z)-	1576-95-0	24.00	0.25	1.98	0.38
17	2-Pentenal, (E)-	1576-87-0	14.33	1.04	27.91	0.00
18	2-Propenal	107-02-8	4.80	0.32	11.78	0.03
19	2(5H)-Furanone, 5-ethyl-	2407-43-4	30.60	0.64	0.26	0.01
20	2,2-Dimethylocta-3,4-dienal	590-71-6	31.33	0.02	0.64	0.59
21	2,3-Butanedione	431-03-8	7.53	1.13	14.74	0.01
22	2,3-Pentanedione	600-14-6	10.67	1.13	29.15	0.00
23	2,4-Decadienal	2363-88-4	34.13	0.36	50.57	0.00
24	2,4-Heptadienal, (E,E)-	4313-03-5	28.33	1.12	109.14	0.01
25	2,4-Hexadienal, (E,E)-	142-83-6	26.93	0.62	0.18	0.01
26	2,7-Nonadien-5-one, 4,6-dimethyl-	74630-80-1	22.93	0.11	0.18	0.02
27	3-Octen-2-one, (E)-	18402-82-9	26.93	0.28	32.4	0.13
28	3,3,5,5-Tetramethylcyclopentene	38667-10-6	36.13	0.06	0.65	0.05
29	3,5-Octadien-2-one, (E,E)-	30086-02-3	29.47	1.03	18.98	0.04
30	4-Heptenal	62238-34-0	36.40	0.26	20.53	0.14
31	4,5-Heptadien-2-one, 3,3,6-trimethyl-	81250-41-1	34.27	0.13	0.19	0.46
32	6-Oxabicyclo [3.1.0]hexan-2-one	6705-52-8	18.27	0.04	1.91	0.42
33	a-Ionone	127-41-3	34.53	0.18	0.17	0.10
34	Acetaldehyde	75-07-0	3.53	0.06	0.96	0.92
35	Acetic acid	64-19-7	27.33	1.77	6.74	0.00
36	Acetic acid, hexyl ester	142-92-7	22.87	0.21	0.06	0.00
37	Acetic acid, methyl ester	79-20-9	4.47	0.29	0.33	0.02
38	Acetoin	513-86-0	22.40	0.99	31.08	0.13
39	Acetone	67-64-1	4.33	1.02	20.21	0.20
40	Acetophenone	98-86-2	31.60	0.20	11.15	0.03
41	Benzaldehyde	100-52-7	29.27	0.79	6.34	0.01
42	Benzaldehyde, 3-ethyl-	34246-54-3	32.53	0.09	0.11	0.00
43	Benzene, 1,4-dichloro-	106-46-7	28.47	0.06	7.73	0.01
44	Benzeneacetaldehyde	122-78-1	31.27	0.54	3.58	0.00
45	Butanal	123-72-8	5.13	0.40	6.97	0.00
46	Butanal, 3-methyl-	590-86-3	5.87	1.13	18.02	0.05
47	Butanoic acid, 3-methyl-	503-74-2	31.27	0.65	1.5	0.22
48	Cyclohexanol, 2,4-dimethyl-	69542-91-2	33.47	0.08	4.01	0.01
49	Dimethyl sulfide	75-18-3	3.80	1.71	30.91	0.00
50	Ethanol	64-17-5	6.27	0.54	0.55	0.30
51	Ethyl Acetate	141-78-6	5.27	0.13	1.86	0.63
52	Furan, 2-ethyl-	3208-16-0	6.67	0.66	3.57	0.00
53	Furan, 2-pentyl-	3777-69-3	19.93	0.89	6.6	0.00
54	Furan, 2,3-dihydro-5-methyl-	1487-15-6	5.60	0.07	6.88	0.15
55	Furfural	98-01-1	27.93	1.90	141.81	0.00
56	Heptanal	111-71-7	17.33	0.48	198.23	0.00
57	Hexanal	66-25-1	11.60	2.52	2.34	0.02
58	Methacrolein	78-85-3	10.20	0.02	0.79	0.85
59	Methanethiol	74-93-1	3.467	0.20	13.24	0.00
60	Methional	3268-49-3	27.67	0.28	11.08	0.00
61	Methyl Alcohol	67-56-1	5.40	2.48	2.52	0.00
62	Methyl formate	107-31-3	3.87	0.12	6.09	0.18
63	n-Caproic acid vinyl ester	3050-69-9	31.20	0.15	2.22	0.02
64	Nonanal	124-19-6	26.67	0.47	5.7	0.05
65	Octanal	124-13-0	23.47	0.36	16.7	0.00
66	Octanoic acid, ethyl ester	106-32-1	28.00	0.06	4.21	0.23
67	Pentanal	110-62-3	7.53	1.48	177.49	0.03
68	Pentanoic acid	109-52-4	33.80	0.28	1.85	0.54
69	Phenol, 4-(1-methylpropyl)-	99-71-8	34.47	0.14	0.05	0.00
70	Propanal	123-38-6	4.07	1.53	44.72	0.00
71	Propanal, 2-methyl-	78-84-2	4.27	0.76	28.8	0.01
72	Propanoic acid	79-09-4	29.13	0.65	30.3	0.00
73	trans-a-Ionone	79-77-6	35.73	0.20	0.13	0.01
74	1,4-Pentadiene	591-93-5	3.47	0.06	3.44	0.01
75	2-Butenal	4170-30-3	10.20	0.73	8.85	0.02
76	2-Penten-1-ol, (E)-	1576-96-1	23.73	0.23	22.72	0.34
77	Acetonitrile	75-05-8	8.40	0.11	4.12	0.01
78	Butyrolactone	96-48-0	31.07	0.85	1837	0.00
79	Ethyl ether	60-29-7	3.33	0.16	51.7	0.00

Note: RT—retention time; VIP—variable importance for the projection.

## Data Availability

The data presented in this study are available on request from the corresponding author.

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
