# Peer review of "Study on Quality Change and Processing Suitability Evaluation of the Low-Temperature Vacuum Frying of Bananas"

_foods, 2023, doi:10.3390/foods12091822_

Round 1

Reviewer 1 Report

The article needs to be refined in many aspects. Linguistic and substantive revision of the article is needed. More information needs to be provided in the methodology and the discussion with the literature and conclusions need to be expanded. Below are my detailed comments:

line 40-43- these sentences require citations. In addition, the authors must explain in the text how the aforementioned studies differ from those conducted and described in this manuscript.

line 67-69- these sentences require citations. In addition, the authors must explain in the text how the aforementioned studies differ from those conducted and described in this manuscript.

What is missing from the introduction is that the quality of the product is also affected by the frying medium, i.e. the type and quality of oil. As research shows, the frying medium is very important in the case of frying protein raw materials as well as those rich in carbohydrates. Here I provide recommended literature: https://doi.org/10.3390/antiox10101637; https://doi.org/10.1016/j.foodchem.2022.13523.

The objective and the scope of the research work were clearly and lucidly defined.

Table 1 - The authors write in the footer of the table that samples 151 and CB5 were not validated, but in a later section of the article these samples are evaluated with the created model for processability. Please verify and correct this.

line 85- Please explain what is Guaranteed Reagent?

line 80, 97- Please explain in the text  what is “ripening stage 5”?

2.3.1. Preparation of fresh banana – please add what for ?!

2.3.2. Preparation of banana slices by low-temperature vacuum frying – instead “by” should be “for”

line 106 - Please explain what is the difference between frying and boiling and what were the temperature and pressure parameters?

line 107- crushed with what? provide conditions

line 110-113- Describe how you analyze the color of banana powder. How is it possible to study hardness of banana powder? Earlier the authors describe that they crushed and sieved the fried banana slices all the samples.

2.6. Determination of nutritional quality index - Describe the procedure of each of the mentioned methodologies with special emphasis on the extraction of the different components being determined. Stating on what device you did the study and citing the literature is insufficient. The methodology should be given so that it can be reproduced. Please check the cited literature because often something else is studied in the cited article or there is a lack of methodology you cite.

In sections 3.1.1 through 3.1.4, there is no discussion of the results with the literature.

Table 7 – please separate three last lines from the rest of them to distinguish indicators names from results of factor analysis

lines 264-278- Please write how the model formula and indicator weights given in Table 9 were generated.

line 269- should be table 9, please check

line 286 - should be table 10, please check

3.5. OPLS-DA of differential flavor substances in banana - Why only 3 samples of banana slices were used for this study? was it 3 types or 3 samples of a selected type? according to me, all 15 types of bananas tested should participate here. When there are 3 samples, the OPLS-DA model is sure to come out very effectively and will nicely arrange the samples before and after frying.

3.7. Identification of the substances affecting the sensory flavor in banana before and after low-369 temperature vacuum frying - this section should be removed and rewritten. The researchers did not perform olfactometry so they don't know what each compound smells like and whether it was detectable in the banana samples tested.

figure 5, 6 - This figures should be removed for the same reason as I stated above. They represent an oversimplification. Researchers have not studied this. These are just guesses.

Conclusions- To the first class also included bananas Guijiao No. 6 and 2. Please provide more conclusions and insights on the evaluation and classification of bananas in terms of suitability for frying because this was the main purpose of the study.

Author Response

Dear Editor and Reviewers:
Thank you for giving us the opportunity to revise our manuscript and your valuable comments concerning our manuscript entitled “Study On Quality Change And Processing Suitability Evaluation Of Low-temperature Vacuum Frying Banana” (ID: foods-2343882). The comments are very helpful for improving our manuscript. We have revised the manuscript carefully according to reviewers’ comments point by point. All modifications to the manuscript are highlighted in red for your further comments. The responses to the reviewer’s comments are as follows

Reviewer 2 Report

Study On Quality Change And Processing Suitability Evaluation Of Low-temperature Vacuum Frying Banana

Line number 34: Mention production statistics and losses associated with this fruit

Line number 42: Elaborate nutritional value/ information and also provide reference. Please check this paper for reference:

The quality behavior of ultrasonically extracted sunflower oil and structural computation of potato strips appertaining to deep frying with thermic variations. Journal of Food Processing and Preservation, 44(10), e14809.

Line number 44: Unfavorable flavor quality is linked with factors, mention these factors

Line number 45. Need reference.

Line number 52: Elaborate on the main findings

Line number 53, 54: Variety, as well as processing factors, are also an important factor in quality evaluation

Line number 57: Volatile metabolites should be mentioned here

Line number 67: Volatile components reduced by which proportion?/ Percentage

2.3: If you followed any protocol then mention its citation

Line number 103: Recheck the statement

Line number 196: Check the statement/ abbreviations and their full form

Line number 205: Cross-check the unit of phenolics

Line number 255: Table 7 Cross-check the term accumulative variance contributio (%)

Vc: Use the full form in Table 7

3.1: Add some cross-references if possible

3.4, 3.7: Add some previous studies/ cross-references

373: Cross-check the statement

The references should be updated

Author Response

(The authors gave the same response as above.)

Round 2

Reviewer 1 Report

The authors addressed all comments and observations. After reading the revised work, I believe that it has been upgraded in quality. Both the introduction, methodology and discussion of the results have been improved.
The authors have also improved the tables and graphs.